# Medication Non-Adherence among Patients with Chronic Diseases in Makkah Region

**DOI:** 10.3390/pharmaceutics14102010

**Published:** 2022-09-22

**Authors:** Majed A. Algarni, Meznah S. Althobiti, Sarah A. Alghamdi, Huriyyah A. Alotaibi, Ohoud S. Almalki, Adnan Alharbi, Mohammad S. Alzahrani

**Affiliations:** 1Department of Clinical Pharmacy, College of Pharmacy, Taif University, P.O. Box 11099, Taif 21944, Saudi Arabia; 2Clinical Pharmacy Department, College of Pharmacy, Umm Al-Qura University, Makkah 21955, Saudi Arabia

**Keywords:** medication adherence, chronic disease, Makkah region, Saudi Arabia

## Abstract

Background: The Makkah region is the most populated region in Saudi Arabia. Studying medication adherence levels may help to improve general health outcomes and decrease overall health care expenditures. Methods: We used the ARMS scale to assess medication adherence. Bivariable analysis of medication non-adherence was performed. Simple and multiple logistic regression models were built to identify factors associated with medication non-adherence. Results: Participants from the Makkah region were more than two times more likely to be non-adherent to their medications compared to other regions (adjusted OR = 2.58, 95% CI: 1.49–4.46). Patients who dispensed their prescriptions at their own expense were two times more likely to be non-adherents (adjusted OR = 2.36, 95% CI: 1.11–4.98). Patients who had a monthly income ≤6000 SR were almost two times more likely to be non-adherents (unadjusted OR = 1.73, 95% CI: 1.05–2.84). Conclusion: Medication adherence is one of the most important factors to help managing the disease. We found that Makkah chronic patients are more likely to be non-adherent with their medications compared to other regions’ patients. Moreover, we found that lower monthly incomes and paying for medications out-of-pocket were significant predictors of medication non-adherence.

## 1. Introduction

Chronic diseases as defined by the Centers for Disease Control and Prevention (CDC) are diseases that last one year or more and need continuing medical care or limit daily activities or both [1]. These conditions are not only the leading cause for disability and death, but they also consume health care and trillions of dollars around the world [1,2]. In its report in 2005, the World Health Organization (WHO) said that around 60% of all deaths in the world are due to chronic diseases [3], and this alarming figure has increased since 2005 to reach 70% in 2019 based on another WHO report [4]. This percentage may increase in the next years as the disease profile changes globally for many reasons due to population age [5]. Managing these chronic diseases usually requires long-term pharmacological treatments. However, many patients do not get the full required benefit of these treatment plans due to the lack of the appropriate medication adherence [6], which may worsen their condition [7]. On the other hand, a high level of medication adherence decreases hospitalization rate and overall health care costs for chronic conditions [8]. The WHO reported that, in developing countries, only half of patients with chronic disease adhere to their treatment plan [9], which means that non-adherence is a real issue that faces many countries around the world.

Many published studies have described and studied medication adherence and its related problems in the Kingdom of Saudi Arabia. For example, in a study that was published in 2019, the researchers concluded that only 7.5% of the surveyed patients were considered as good adherents to their hypertensive medications [10]. Another study, conducted in Makkah city, showed that the majority of the patients did not meet their treatment goal, and that was mainly because of the poor medication adherence for those patients [11]. In a study that was conducted on patients with type 2 diabetes mellitus, the authors concluded that only 35% of the surveyed patients were in a high level of adherence to anti-diabetic drugs [12]. In another study conducted on hypertensive patients attending primary care clinics, the researchers found that more than half of the patients (54%) were non-adherent to their medications [13]. A published study in 2016, that was conducted in Riyadh city on cardiac patients found that only a quarter of the patients (24.5%) were in high-level adherence [14]. A study that was conducted on multiple sclerosis patients in Saudi Arabia found that only around 10% were adherent to their medications [15]. Another study conducted on diabetic patients type 2 in Alkharj city found that only 26.5% of the patients were in high-level adherence [16]. Another study conducted in Al-Qassim region on diabetic and hypertensive patients showed that 52% of the patients reported low-adherence level [17]. A study conducted in the Aseer region on heart failure patients found that more than half of the patients had a poor adherence to their medications [18]. In another study conducted in Taif city on patients on oral anticoagulants drugs, the authors stated that the adherence rate to oral anticoagulants was 35.9% [19]. Moreover, in another study that was conducted on diabetic patients, the researcher found that non-adherence to medications was 80% [20].

Based on the general authority for statistics in Saudi Arabia, the Makkah region is the most populated region in the kingdom, with 33.28% of the population living there [21]. This large portion of the population suggests that the Makkah region also consumes the highest percentage of the health care budget, and that a considerable percentage of patients with chronic conditions live in this region.

This study aims to estimate the medication adherence level across patients with chronic conditions, and to understand the factors attributed to medication non-adherence.

Studying the medication adherence level among chronic patients in this highly populated region may help to improve general health outcomes, and it may also help to decrease overall health care expenditures by improving the quality of life of chronic patients.

## 2. Materials and Methods

### 2.1. Settings and Participants

A cross-sectional study was conducted using a self-reported survey to assess medication adherence for patients with chronic conditions. A convenience sampling strategy was used to recruit patients with chronic diseases living in the Makkah region, from December 2021 to March 2022. The self-reported survey was distributed using both direct distribution and electronic distribution. The eligible study population included those who are 18 years or older and diagnosed with at least one chronic condition. All participants were provided with informed consent, and the identities of the respondents were not revealed. An approval from the Scientific Research Ethics Committee at Taif University was obtained.

### 2.2. Survey Tool and Data Collection

The survey was divided into two sections. The first section was about the sociodemographic and clinical characteristics, including gender, whether the patient lives with family or alone, education level, monthly income, number, and type of chronic diseases, whether the patients dispensed their medication in a hospital or a clinic, and whether the patients are paying for their medications out-of-pocket. The second section of the survey was the translated version of the Adherence to Refills and Medications Scale (ARMS) [22]. ARMS is a 12-item medication adherence assessment tool with established validity and reliability in English and was published in 2009 by Dr. Kripalani and his colleagues [22]. It was translated and validated after that into many languages among patients with chronic health conditions [23]. It was translated and validated recently to the Arabic language among Saudi patients [23], which makes it a good and suitable survey for our study. This scale could be easily completed by patients in a short time (less than 15 min).

ARMS involves two subscales. The first one is an adherence with filling medications, while the second one is an adherence with taking medications. The first subscale consists of four items, while the second subscale contains eight items. Each one of the items is scored using a 4-point Likert-scale (none = 1 point, some = 2 points, most = 3 points, All = 4 points). ARMS scores could range from 12 points to 48 points, with higher scores suggesting poor adherence [22]. We used a score of 16 as a cut-off point, where patients with 16 points or more were categorized as non-adherent, and patients with less than 16 points were categorized as adherent, as suggested by a previous study [24].

### 2.3. Statistical Analysis

Descriptive analysis of sociodemographic characteristics was conducted. Median scores with interquartile range were calculated for the total ARMS score. Frequencies and percentages were calculated for categorical variables. Bivariable analysis of medication non-adherence was performed using Pearson’s Chi square test or Fisher’s exact test where appropriate. Simple and multiple logistic regression models were built to identify factors associated with medication non-adherence. Odds ratios [OR] with 95% confidence intervals (CIs) were calculated and used to determine the magnitude of associations. *p*-values of less than 0.05 were considered statistically significant. All analyses were performed using SAS University Edition (SAS Institute, Cary, NC, USA).

## 3. Results

A total of 498 patients responded to the survey. Table 1 shows the patients’ sociodemographic and clinical characteristics. The majority were female participants (66.7%). The participants from the Makkah region represented more than half of the participants with 263 participants (52.8%), while those from other regions were 235 participants (47.2%). Out of the 498 participants, 242 (48.6%) were at the age of 40 or younger, 109 participants (21.9%) were between the ages of 41 and 50, and 83 participants (16.7%) were between the ages of 51 and 60. Only 64 participants (12.9%) were over the age of 60.

Almost half of the participants (52.6%) had high school or lower education. Most of the participants (62.7%) had a monthly income ≤6000 Saudi Riyal.

In all, 308 participants (61.9%) dispensed their medication from governmental pharmacies, while 190 participants (38.1%) dispensed their medication from private pharmacies. Approximately two-thirds of the participants (68.8%) had one chronic condition, 85 participants (17%) had two chronic conditions, while 70 participants (14.1%) had three or more chronic conditions.

The most frequent chronic condition reported was diabetes (44.4%), followed by rheumatoid arthritis (25.7%) and hypertension (25.5%). More than one-third (38%) had to pay on their own expense to dispense their medication, with no governmental or insurance coverage.

Patients from both the Makkah region and other regions were similar in all characteristics, except for age (*p* = 0.034) and ARMS score (*p* = 0.016) (Table 1).

As shown in Table 2, only 74 participants (15%) had an ARMS score of less than 16 points and were considered as the high-adherent group. Those with an ARMS score of 16 points or more (85%) were considered as the low-adherent group. Most of the participants who live in the Makkah region (91%) are among the low-adherence level group, and only 9% were among the high-adherence level group. On the other hand, 21% of the participants who live in other regions were among the high-adherence level group.

The logistic regression analysis of non-adherence is shown in Table 3. Participants from the Makkah region were more than two times more likely to be non-adherent to their medications compared to other regions (adjusted OR = 2.58, 95% CI: 1.49–4.46). Moreover, patients who dispensed their prescriptions at their own expense were two times more likely to be non-adherents compared to patients who had insurance or governmental coverage (adjusted OR = 2.36, 95% CI: 1.11–4.98). Patients who had a monthly income of less than or equal to 6000 SR were almost two times more likely to be non-adherents compared to those who had more than 6000 SR (unadjusted OR = 1.73, 95% CI: 1.05–2.84).

## 4. Discussion

The goal of this study was to assess medication adherence among patients with chronic conditions in the Makkah region. Our study revealed that the majority of patients with chronic conditions (85%) have low adherence to prescribed medications. Our findings fall in line with earlier studies conducted in Saudi Arabia, which also reported a low level of adherence in populations with different chronic conditions [11,12,13,14,17,18,19,25]. It is not easy for patients to maintain high-level medication adherence, especially those with chronic conditions, and their adherence could be affected by a variety of factors. There are conflicting viewpoints and results about those factors that influence medication adherence. In this study, the level of adherence was significantly associated with geographic region, monthly income, and paying for medicines out-of-pocket.

In terms of geographic region, our findings showed that the area of residence was significantly associated with adherence level. Compared to other regions, participants from the Makkah region were more than two times more likely to be non-adherent to their medications. Another study carried out in Makkah also revealed that the patients had a lower level of adherence [11], which is consistent with the findings of our study. Being the most populated region in Saudi Arabia, with a higher number of foreigners and expatriates [21,26], may have contributed to non-adherence in Makkah region, given that the majority of expatriates are among the low monthly income class.

We found that monthly income had an impact on adherence levels. Patients with lower incomes are more likely to have a lower adherence level. One explanation could be that patients with lower incomes may consider it to be a secondary issue compared to other issues in their life, or it could simply be that they cannot afford to pay for their medications. Studies have shown that patients without insurance or with low income show less treatment adherence [18].

We also found that patients who had their prescriptions filled at their own expense have lower adherence compared to those who have insurance or governmental coverage. As we mentioned above, the expense of medications was a barrier to adherence, which may have contributed to this result.

Our study found that there were no statistically significant differences regarding patients’ adherence according to their gender, age, education status, marital status, site of prescription dispensing, or number of chronic conditions.

In terms of educational level, we had anticipated that patients with higher levels of education would have more health information and, as a result, exhibit better adherence. Surprisingly, our results revealed no association between educational status and drug adherence. Similar findings were found by another study conducted in Saudi Arabia, which indicated that patients with and without education had an equal likelihood of adhering to their treatment regimen or not [12]. According to another study conducted in Saudi Arabia, patients with low literacy levels have less of a tendency to be adherent [17], because they may be unaware of possible concerns and side effects of their medications [27]. However, a couple of other studies conducted in Saudi Arabia revealed that higher education was negatively correlated with adherence. The reason could be that those with higher education were more likely to be employed and had more responsibilities in their lives, which made it difficult for them to remember to take their medication regularly [15,18]. Other studies have shown a correlation between higher levels of education and a higher compliance rate [28]. Due to these inconsistent findings, it is possible that education is not a reliable indicator of therapeutic compliance [28].

We found no association between gender, age, and medication non-adherence. A previous study found that females were more compliant [28], while other studies found that males are more compliant [13,29]. There have been conflicting findings about the relationship between age and level of adherence. Some studies have found that patients who are older tend to adhere to their treatment regimens better [30].

There are a number of limitations in this study, including its cross-sectional design, which may have under- or over-reported the true incidence of patients’ poor adherence. Additionally, there are additional factors that could influence adherence but were not examined in this study, such as the number of medications, patients’ perceptions of their illness and treatment, physical and mental capabilities [13], the number of children, disease severity, and the number of tablets used daily [31].

## 5. Conclusions

Medication adherence for chronic patients is one of the most important factors to help manage disease. This study found that chronic patients in Makkah are more likely to be non-adherent with their medications compared to patients in other regions. Additionally, we found that lower monthly incomes and paying for medications out-of-pocket were significant predictors of medication non-adherence.

## Figures and Tables

**Table 1 pharmaceutics-14-02010-t001:** Sociodemographic and clinical characteristics of study patients.

Characteristic	Overall *n* = 498	Geographical Region		
		Makkah Region *n* = 263	Other Regions *n* = 235	*p*
**Sex**, Female	332 (66.7)	182 (69.2)	150 (63.8)	0.204
**Age groups,**				0.034
≤40 years	242 (48.6)	127 (48.3)	115 (48.9)	
41–50 years	109 (21.9)	52 (19.8)	57 (24.3)	
51–60 years	83 (16.7)	55 (20.9)	28 (11.9)	
>60 years	64 (12.9)	29 (11)	35 (14.9)	
**Level of education,**				0.233
High school or less	262 (52.6)	118 (44.9)	117 (49.7)	
Bachelor or higher education	236 (47.4)	145 (55.1)	118 (50.2)	
**Marital status**, married	368 (73.9)	195 (74.1)	173 (73.6)	0.893
**Monthly income,**				0.086
≤6000 SR	312 (62.7)	174 (66.2)	138 (58.7)	
>6000 SR	186 (37.3)	89 (33.8)	97 (41.2)	
**Site of prescription dispensing,**				0.174
Governmental	308 (61.9)	170 (64.6)	138 (58.7)	
Private or Comm. Pharmacy	190 (38.1)	93 (35.4)	97 (41.2)	
**No. of chronic conditions,**				0.273
One	343 (68.8)	178 (67.7)	165 (70.2)	
Two	85 (17.1)	42 (15.9)	43 (18.3)	
Three or more	70 (14.1)	43 (16.4)	27 (11.5)	
**Type of chronic conditions**, †				
DM	221 (44.4)	126 (47.9)	95 (40.4)	0.093
Rheumatoid arthritis	128 (25.7)	62 (23.5)	66 (28.1)	0.251
Hypertension	126 (25.3)	76 (28.9)	50 (21.3)	0.051
Dyslipidemia	77 (15.5)	45 (17.1)	32 (13.6)	0.282
CVD	57 (11.5)	37 (14.1)	20 (8.5)	0.052
Others (e.g., asthma, anemia…)	142 (28.5)	66 (25.1)	76 (32.3)	0.144
**Out-of-pocket prescriptions**, yes	190 (38.2)	104 (39.5)	86 (36.6)	0.499
**ARMS** **total score**	20 [17,18,19,20,21,22,23,24]	20 [18,19,20,21,22,23,24,25]	19 [16,17,18,19,20,21,22,23,24]	0.016

Values are reported as number (%) or median [interquartile ranges]. Abbreviations: DM, diabetes mellitus; CVD, cardiovascular disease; ARMS, the Adherence to Refills and Medications Scale. † Types of chronic conditions do not add up to 100% as some patients have more than one condition.

**Table 2 pharmaceutics-14-02010-t002:** Characteristics of patients according to their levels of adherence.

Characteristic	Adherence Levels		
	High (ARMS < 16), *n* = 74	Low (ARMS ≥ 16), *n* = 424	*p **
**Gender,** *n* (%)			0.154
Male	30 (18.1)	136 (81.9)	
Female	44 (13.3)	288 (86.7)	
**Age group,** *n* (%)			0.353
≤40 years	29 (11.9)	213 (88.1)	
41–50 years	18 (16.5)	91 (83.5)	
51–60 years	15 (18.1)	68 (81.9)	
>60 years	12 (18.8)	52 (81.2)	
**Level of education,** *n* (%)			0.626
High school or less	37 (14.1)	225 (85.9)	
Bachelor/higher education	37 (15.7)	199 (84.3)	
**Geographical region,** *n* (%)			**<0.001**
Makkah region	24 (9.1)	239 (90.9)	
Other regions	50 (21.3)	185 (78.7)	
**Married,** *n* (%)			0.567
No	17 (13.1)	113(86.9)	
Yes	57 (15.5)	311 (84.5)	
**Monthly income,** *n* (%)			**0.029**
≤6000 SR	38 (12.2)	274 (87.8)	
>6000 SR	36 (19.4)	150 (80.6)	
**Site of prescription dispensing,** *n* (%)			0.842
Governmental	45 (14.6)	263 (85.4)	
Private or Comm. Pharmacy	29 (15.3)	161 (84.7)	
**No. of chronic conditions,***n* (%)			0.556
One	47 (13.7)	296 (86.3)	
Two	15 (17.6)	70 (82.4)	
Three or more	12 (17.1)	58 (82.9)	
**Out-of-pocket prescriptions,** *n* (%)			**0.037**
No	54 (17.5)	254 (82.5)	
Yes	20 (10.5)	170 (89.5)	

* *p*-values produced using Pearson’s *Chi* square or Fisher’s exact test.

**Table 3 pharmaceutics-14-02010-t003:** Binary logistic regression analysis of non-adherence (ARMS total score < 16).

Factors	Unadjusted OR (95% CI)	*p*	Adjusted OR (95% CI)	*p*
**Gender,**				
Male	Ref		Ref	
Female	1.44 (0.87–2.39)	0.155	1.07 (0.62–1.87)	0.804
**Age group,**				
>60 years	Ref		Ref	
51–60 years	1.05 (0.45–2.42)	0.573 0.903	0.85 (0.35–2.07)	0.173
41–50 years	1.17 (0.52–2.61)	0.074	1.1 (0.45–2.69)	0.314
≤40 years	1.69 (0.81–3.54)		1.61 (0.64–4.06)	0.144
**Level of education,**				
Bachelor/higher education	Ref		Ref	
High school or less	1.13 (0.69–1.85)	0.626	1.05 (0.58–1.91)	0.873
**Geographical region,**				
Other regions	Ref		Ref	
Makkah region	2.69 (1.59–4.54)	**<0.001**	2.58 (1.49–4.46)	**0.001**
**Married,**				
No	Ref		Ref	
Yes	0.82 (0.45–1.47)	0.506	1.28 (0.63–2.62)	0.499
**Monthly income,**				
>6000 SR	Ref		Ref	
≤6000 SR	1.73 (1.05–2.84)	**0.031**	1.53 (0.85–2.74)	0.158
**Site of prescription dispensing,**				
Private or Comm.	Ref		Ref	
Governmental	1.05 (0.64–1.75)	0.842	1.63 (0.81–3.31)	0.174
**Number of chronic conditions**				
One	Ref		Ref	
Two	0.74 (0.39–1.40)	0.747	0.91 (0.45–1.84)	0.963
Three or more	0.77 (0.38–1.53)	0.617	0.85 (0.38–1.87)	0.764
**Out-of-pocket prescriptions,**				
No	Ref		Ref	
Yes	1.81 (1.04–3.12)	**0.034**	2.36 (1.11–4.98)	**0.025**

## Data Availability

Data are available upon reasonable request to the corresponding author.

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
