# Peer review of "Medication Non-Adherence among Patients with Chronic Diseases in Makkah Region"

_pharmaceutics, 2022, doi:10.3390/pharmaceutics14102010_

Round 1

Reviewer 1 Report

The manuscript used a translated and validated questionnaire ARMS to evaluate adherence in Makkah region and other regions of Saudi Arabia.

ln 99 - please say the time of completing

Insert one more column in Table 1 with the p-value from a comparison of the two regions. And discuss the results.

You did not say anything about the responses to the ARMS questions. What were the factors decreasing the adherence from the responses to the ARMS items? More unintentional or intentional factors? These tools of measuring adherence (doi:10.3390/pharmaceutics13101683. https://pubmed.ncbi.nlm.nih.gov/34683976/) are very important in assessing how some interventions could enhance adherence.

Author Response

Dear reviewer,

Thanks for your suggestions.

We added the time pf completing as you suggested in line 99.

We also inserted p-value column in Table 1 as you have suggested and discussed the results.

Reviewer 2 Report

The aim of this study is to estimate the medication adherence level across patients with 71 chronic conditions, and to understand the factors attributed to medication non-adherence. This topic is important and fits with Pharmaceutics journal. Several suggestions are provided for the authors’ reference.

1.      Title: The authors may consider whether ‘Non-adherence’ or ‘Adherence’ better fits with research goal.

2.      Materials and methods:

1)          How participants are recruited for direct distribution and electronic distribution should be clearly explained. Further, the homogeneity of these two types of participants should be examined.

2)          How many responses were collected, how many responses were excluded, what are the excluding reasons should be explained as well.

3)          The time span for data collection should be articulated.

3.      Results

1)          Factor analysis should be conducted to ensure the validity for ARMS.

2)          The unadjusted OR of monthly income is 1.73, is it appropriate to express in ‘two times?’ Will ‘near two times’ be more appropriate?

3)          Table 1: The last row, ARMS total score, is it the average ARMS score?

4.      Discussion

1)          Second paragraph about geographic region, the authors may consider explicitly explain why foreigners and expatriates may contribute to non-adherence of medication in Makkah region.

2)          Non-significant factors and their respective potential reasons should be discussed as well.

Author Response

Dear reviewer,

Thanks for your constructive suggestions.

We have corrected the title to “non-Adherence” as you have implied, it fits better in the title.

Regarding the recruitment of the participants, we planned for both direct and electronic recruitment. However, we faced a problem with direct recruitment inside clinics and hospitals, so we preferred to continue with electronic recruitment, and only 8 participants were recruited through direct recruitment. So, there is no need to examine the homogeneity between the two groups. 

Also, you suggested that it is better to clarify the reasons for excluded participants. There were no excluded participants, as we explained in the manuscript, the eligibility criteria included those who are 18 years or older and having at least one chronic condition. So, once a participant did not fit with those criteria, he or she will be not eligible and cannot continue the electronic survey.

The time span for the survey data collection was from December 2021 to March 2022, as it was mentioned in the manuscript, in line 81,82.

You asked also for a factor analysis to ensure the validity of ARMS survey, but as we mentioned in the manuscript in the Survey Tool and Data Collection section “It was translated and validated recently to Arabic language among Saudi patients, which makes it a good and suitable survey for our study” (line 96-98).

We have corrected the expression of “1.73” OR, to become “almost” two times.

You asked about the last row in Table 1, the values represent the median [interquartile range], as mentioned in the table legend.

We also added the reason why the high number of expatriates may increase the non-adherence in Makkah region compared to other regions. (line 175-178)

You asked for non-significant factors to be discussed, we added a whole paragraph for those non-significant factors. (lines 208-212) We also have already discussed non-significant factors in the previous non revised manuscript such as education level. (195 -207)